# Analysis of *Carica papaya* Informs Lineage-Specific Evolution of the Aquaporin (AQP) Family in Brassicales

**DOI:** 10.3390/plants12223847

**Published:** 2023-11-14

**Authors:** Zhi Zou, Yujiao Zheng, Zhengnan Xie

**Affiliations:** Hainan Key Laboratory for Biosafety Monitoring and Molecular Breeding in Off-Season Reproduction Regions, Institute of Tropical Biosciences and Biotechnology/Sanya Research Institute of Chinese Academy of Tropical Agricultural Sciences, Haikou 571101, China; zhengyujiaosusie@163.com (Y.Z.); xiezhengnan@itbb.org.cn (Z.X.)

**Keywords:** Caricaceae, Brassicaceae, orthogroup, synteny, fruit development

## Abstract

Aquaporins (AQPs), a type of intrinsic membrane proteins that transport water and small solutes across biological membranes, play crucial roles in plant growth and development. This study presents a first genome-wide identification and comparative analysis of the *AQP* gene family in papaya (*Carica papaya* L.), an economically and nutritionally important fruit tree of tropical and subtropical regions. A total of 29 *CpAQP* genes were identified, which represent five subfamilies, i.e., nine plasma intrinsic membrane proteins (PIPs), eight tonoplast intrinsic proteins (TIPs), seven NOD26-like intrinsic proteins (NIPs), two X intrinsic proteins (XIPs), and three small basic intrinsic proteins (SIPs). Although the family is smaller than the 35 members reported in Arabidopsis, it is highly diverse, and the presence of *CpXIP* genes as well as orthologs in *Moringa oleifera* and *Bretschneidera sinensis* implies that the complete loss of the XIP subfamily in Arabidopsis is lineage-specific, sometime after its split with papaya but before Brassicaceae–Cleomaceae divergence. Reciprocal best hit-based sequence comparison of 530 AQPs and synteny analyses revealed that *CpAQP* genes belong to 29 out of 61 identified orthogroups, and lineage-specific evolution was frequently observed in Brassicales. Significantly, the well-characterized NIP3 group was completely lost; lineage-specific loss of the NIP8 group in Brassicaceae occurred sometime before the divergence with Cleomaceae, and lineage-specific loss of NIP2 and SIP3 groups in Brassicaceae occurred sometime after the split with Cleomaceae. In contrast to a predominant role of recent whole-genome duplications (WGDs) on the family expansion in *B. sinensis*, *Tarenaya hassleriana*, and Brassicaceae plants, no recent *AQP* repeats were identified in papaya, and ancient WGD repeats are mainly confined to the PIP subfamily. Subfamily even group-specific evolution was uncovered via comparing exon–intron structures, conserved motifs, the aromatic/arginine selectivity filter, and gene expression profiles. Moreover, down-regulation during fruit ripening and expression divergence of duplicated *CpAQP* genes were frequently observed in papaya. These findings will not only improve our knowledge on lineage-specific family evolution in Brassicales, but also provide valuable information for further studies of *AQP* genes in papaya and species beyond.

## 1. Introduction

Papaya (*Carica papaya* L., *2n* = 18), which is more likely to origin in South Mexico and/or Central America, is an economically and nutritionally important tree fruit crop widely cultivated in tropical and subtropical areas [1,2]. Papaya is sweet, flavorful, brightly colored, and uniquely rich in vitamin C and carotenoids, making it rank first on nutritional scores among 38 common fruits and also rank first among fruits consumed [3,4]. Papaya also has valuable medical and industrial applications, including the production of a well-known proteolytic enzyme, papain [5,6,7,8,9]. Water balance is particularly important for this special species because a large amount of water is essential for the crop yield and quality [10,11]. Papaya is a member of the Caricaceae family within Brassicales, the same order as the well-known model plant Arabidopsis (*Arabidopsis thaliana*), which may share a common ancestor from about 72 million years ago (Mya) [2,11,12]. The papaya genome was estimated to be 372.0 Mb by using flow cytometry [13], and the first draft genome was reported in 2008, which is derived from SunUp, the first commercial virus-resistant transgenic variety [3]. This assembly spans about 370.4 Mb fragmented in 17,766 scaffolds (Scfs) [3]. More recently, chromosomal-level assemblies were also described for SunUp and its progenitor Sunset, spanning 351.5 and 350.3 Mb in nine chromosomes (Chrs), respectively [14]. Although papaya harbors a considerably bigger genome size than Arabidopsis (more than twofold), its predicted protein-coding genes of 22,394 (SunUp)/22,416 (Sunset) are smaller than 27,655 present in Arabidopsis (Araport11), implying lineage-specific gene evolution and reflecting the occurrence of two additional whole-genome duplications (WGDs, known as At-β and At-α) followed by huge chromosomal rearrangement and massive gene loss occurring in Arabidopsis after the split with papaya [14,15,16].

Aquaporins (AQPs), a group of widely found intrinsic membrane proteins transporting water and/or small solutes across biological membranes, play crucial roles in plant growth, development, and stress responses [17,18]. AQPs belong to the ancient major intrinsic protein (MIP) superfamily, and usually share several typical structural characteristics, including the presence of six transmembrane helices (i.e., TM1–TM6) connected by five loops (i.e., LA–LE), two highly conserved Asn-Pro-Ala (NPA) motifs as well as the so-called aromatic/arginine (ar/R) selectivity filter (i.e., H2 at the TM2, H5 at the TM5, LE1 and LE2 at the LE) [17,19,20]. Whereas dual NPA motifs act as a size barrier of the pore via creating an electrostatic repulsion of protons, the ar/R filter determines the substrate specificity via rendering the pore constriction site diverse in both size and hydrophobicity [19,21,22]. Additionally, an H residue corresponding to H^131^ at the LC of AtTIP2;1 was proven to be essential for NH_3_ permeability [23], whereas a T residue corresponding to T^109^ at the TM1 of rice (*Oryza sativa*) Lsi1 (or OsNIP2;1) as well as the NPA spacing of 108 amino acids (AA) were shown to be essential for silicon transport [24,25]. In the algae kingdom, AQPs are present as one or a few copies. However, more than 19 family members were found in terrestrial plants [26,27,28,29,30]. Moreover, according to sequence similarity, AQPs identified in land plant lineages were clustered into seven evolutionary subfamilies including plasma intrinsic membrane protein (PIP), tonoplast intrinsic protein (TIP), NOD26-like intrinsic protein (NIP), X intrinsic protein (XIP), small basic intrinsic protein (SIP), GlpF-like intrinsic protein (GIP), and hybrid intrinsic protein (HIP) [27,29]. Among them, GIPs, which may be obtained through horizontal gene transfer, have been completely lost in spikemoss (*Selaginella moellendorffii*) and vascular plants, whereas HIPs were only reported in moss (*Physcomitrella patens*) and spikemoss [26,27,28]. Interestingly, the widely distributed XIPs were shown to be absent from monocots and Brassicaceae species including Arabidopsis [29,31,32]. Genome-wide comparison also revealed a key role of recent WGDs on the expansion of the *AQP* gene family [29,30,32,33]. Among 24 *AQP* repeats identified in poplar (*Populus trichocarpa*), 20 were shown to arise from the Salicaceae-specific p WGD [33,34]. In cassava (*Manihot esculenta*), 13 out of 14 identified *AQP* repeats were shown to arise from the recent ρ WGD that was shared by rubber tree (*Hevea brasiliensis*) [30]. In Arabidopsis, 10 out of 17 identified *AQP* repeats were shown to result from two recent WGDs [35]. Nevertheless, whether XIPs are present in other Brassicales families beyond Brassicaceae and lineage-specific evolution patterns of the whole gene family in Brassicales still need to be studied.

In this paper, we present a genome-wide identification and analysis of *AQP* family genes in papaya, including gene structures, evolutionary relationships, sequence characteristics, gene expression profiles, as well as a comprehensive comparison with the basal eudicot *Aquilegia coerulea* (Ranunculaceae, Ranunculales), poplar (Salicaceae, Malpighiales), castor bean (*Ricinus communis*, Euphorbiaceae, Malpighiales), physic nut (*Jatropha curcas*, Euphorbiaceae), cassava (Euphorbiaceae), rubber tree (Euphorbiaceae), and several representative Brassicales species, i.e., horseradish (*Moringa oleifera*, Moringaceae), *Bretschneidera sinensis* (Akaniaceae), spider flower (*Tarenaya hassleriana*, Cleomaceae), saltwater cress (*Eutrema salsugineum*, Brassicaceae), *A. halleri* (Brassicaceae), *A. lyrata* (Brassicaceae), and Arabidopsis (Brassicaceae). Significantly, the presence of XIPs in papaya, horseradish, and *B. sinensis* and their absence from spider flower imply that their loss in Arabidopsis is lineage-specific, occurring sometime after the split with papaya but before Brassicaceae–Cleomaceae divergence. These findings will facilitate further functional studies in papaya and other species.

## 2. Results

### 2.1. Identification, Evolutionary Analysis, and Evolution Patterns of 29 AQP Family Genes in Papaya

As shown in Table 1, the search of papaya genome sequences resulted in 29 loci encoding *AQP* genes from both Sunset and SunUp (ASGPBv0.4). In Sunset, these genes were shown to unevenly distribute across nine chromosomess, varying from a single one of Chr3/-7 to six of Chr2 (Figure 1), in contrast to interspersing among 26 scaffolds in SunUp (Table 1). To facilitate synteny analysis, *CpAQP* genes identified in Sunset were used for further analyses.

To uncover their evolutionary relationships, an unrooted evolutionary tree was constructed using full-length CpAQP proteins together with published AQPs, i.e., 35 AtAQPs, 37 RcAQPs, 31 JcAQPs, 42 MeAQPs, 48 HbAQPs, and 55 PtAQPs. As shown in Figure 2, these AQPs were clustered into five subfamilies, i.e., PIP, TIP, NIP, SIP, and XIP. Moreover, each subfamily could be divided into two to eight groups, i.e., PIP1–2, TIP1–6, NIP1–8, SIP1–3, and XIP1–3. Interestingly, despite the absence of the XIP subfamily in Brassicaceae plants, two XIPs, which belong to the XIP1 and XIP2 groups, respectively, were identified in papaya. Additionally, the NIP2 group, which was not identified in Arabidopsis, is also present in papaya as well as all of the Malpighiales species compared in this study (Figure 2 and Appendix A). By contrast, no NIP3 homolog was found in either papaya or Arabidopsis, though two evolutionary subgroups were identified in several Euphorbiaceae species, e.g., physic nut and cassava. Compared with previous studies, three novel groups, denoted as TIP6, NIP8, and SIP3, were proposed in this study, which only share 67.6/72.8, 60.6/57.6, and 58.2/50.0% sequence identities with their closest homologs (i.e., TIP2, NIP4, and SIP1) at the nucleotide and protein levels, respectively (Figure 2, Table 1 and Appendix A), implying their early divergence. Notably, no SIP3 homolog was found in Arabidopsis, though it is broadly present in other tested species (Figure 2 and Appendix A).

Deduced CpAQP peptides consisted of 236–322 AA with an MW value of 25.09–34.73 kDa, a GRAVY value of 0.327–1.063, and an AI value of 92.83–119.27, which is consistent with their amphipathic feature. Except for CpTIP5;1 (8.71), other TIPs were shown to be acidic with a pI value of less than 7. By contrast, members of other four subfamilies are usually basic with the pI value varying from 7.63 to 10.42, though CpNIP4;1 possesses a small pI value of 5.35. Except for CpNIP8;1 (47.11), all other peptides are likely stable with the II value of less than 40.00 (Table 1). Despite sharing one conserved MIP domain and six TMs connected by five loops (Appendix A, Appendix A and Appendix A), the sequence similarity between different CpAQP family members varies from 20.1% to 95.8% (Appendix A). In accordance with the evolutionary analysis, the SIP subfamily is distant, exhibiting 23.7–30.0%, 28.4–39.0%, 22.4–28.3%, and 20.1–27.3% sequence similarities with the PIP, TIP, NIP, and XIP subfamilies, respectively; the XIP subfamily shares 31.2–38.9%, 29.2–36.6%, and 25.2–37.6% similarities with the PIP, TIP, and NIP subfamilies, respectively, supporting its close relationship with the PIP subfamily. Compared with other subfamilies, higher sequence similarities were observed within PIP subfamily members, ranging from 65.4% to 95.8%, implying their highly conserved evolution. On the contrary, the NIP and TIP subfamilies are more diverse, having evolved into eight and six evolutionary groups with sequence similarities of 43.1–74.5% and 55.9–85.6%, respectively (Figure 2 and Appendix A). Nevertheless, in most cases, the sequence similarities within subfamilies are somewhat higher than those between subfamilies (Appendix A).

According to synteny analysis, seven duplicate pairs, i.e., *CpPIP1;1*/-*1;2*, *CpPIP1;3*/-*1;4*, *CpPIP2;1*/-*2;2*/-*2;3*, *CpPIP2;4*/-*2;5*, *CpTIP1;2*/-*1;3*, and *CpNIP4;1*/*-8;1*, which exhibit 69.2–95.8% similarities at the protein level, were defined as WGD repeats for their location within syntenic blocks (Appendix A). Additionally, *CpXIP1;1* and -*2;1*, which exhibit 47.8% similarity at the protein level, were characterized as tandem repeats for their neighboring location with the interval of only 691 bp; *CpTIP2;1*/*-1;1*, *CpTIP5;1*/*-1;1*, *CpTIP3;1*/*-1;2*, *CpNIP2;1*/*-1;1*, *CpNIP4;1*/*-1;1*, *CpNIP5;1*/*-6;1*, and *CpSIP1;1*/*-2;1*, which exhibit 46.3–73.4% similarities at the protein level, were characterized as transposed repeats; *CpTIP6;1*/*-2;1*, *CpTIP1;2*/*-1;1*, and *CpSIP3;1*/*-1;1*, which exhibit 85.6%, 89.7%, and 63.7% similarities at the protein level, were characterized as dispersed repeats (Appendix A and Figure 1). According to Ks values, these WGD repeats could be divided into two groups, and the young group that includes *CpPIP1;1*/-*1;2*, *CpPIP1;3*/-*1;4*, *CpPIP2;2*/-*2;3*, and *CpPIP2;4*/-*2;5* may arise from the γ whole-genome triplication (WGT) event. Although the dispersed duplicate pair *CpTIP6;1*/*-2;1* exhibits a comparative Ks value to that of the young WGD group (i.e., 1.8799 vs. 1.1851–1.8078), they are more likely to be generated sometime before the split of basal and core eudicots since their orthologs have already been present in *A. coerulea* (see below). Two other dispersed duplicate pairs, i.e., *CpTIP1;2*/*-1;1* and *CpSIP3;1*/*-1;1*, seem to be older, and share similar Ks values with the old WGD group. Additionally, except for *CpNIP4;1* and *-1;1*, the majority of transposed repeats possess considerably high Ks values, implying their early origin and fast evolution. Without any exception, the Ka/Ks ratios of all duplicate pairs are below one (from 0.0258 to 0.3751) (Table 2).

### 2.2. Identification of AQP Genes in Representative Plant Species and Insight into Lineage-Specific Family Evolution in Brassicales

The presence of NIP2, SIP3, XIP1, and XIP2 groups in papaya implies that their loss in Arabidopsis is lineage-specific, and may occur sometime after the Arabidopsis–papaya divergence. However, despite the wide presence of the NIP3 group in Malpighiales plants, no homolog was detected in either papaya or Arabidopsis, implying that lineage-specific loss occurred sometime before the Arabidopsis–papaya divergence. To learn more about lineage-specific family evolution in Brassicales, *AQP* genes were also identified from *A. coerulea*, horseradish, *B. sinensis*, spider flower, saltwater cress, *A. lyrata*, and *A. halleri*. As a basal eudicot, *A. coerulea* was shown to encode 29 *AQP* genes, the same as papaya (Appendix A). Among them, AcTIP7;1, a dispersed repeat of AcTIP6;1, was named for clustering with AcTIP5;1 (Appendix A), possessing the similar ar/R filter (see below) but sharing lower sequence similarity with AcTIP2;1 and AcTIP6;1 (i.e., 62.5% and 63.1% vs. 63.4%). In horseradish, a Moringaceae plant within Brassicales, a total of 28 *AQP* family genes were identified, which is comparative to that of papaya. Nevertheless, one recent tandem repeat was found in the XIP2 group and no orthologs were identified for either *CpNIP8;1* or *CpSIP3;1* in this species (see below). In *B. sinensis*, an Akaniaceae plant within Brassicales, a high number of 53 *AQP* genes (also including six other pseudogenes) were identified, nearly twice that of papaya and reflecting the occurrence of one independent recent WGD. Significantly, one *XIP1*, two *XIP2s*, two *SIP3s*, and one *NIP8* were identified and the NIP5 group has extensively expanded in this species (Appendix A and Appendix A). In spider flower, which belongs to the Brassicaceae sister family Cleomaceae and experienced one independent recent WGT termed Th-α, a total of 36 *AQP* genes were identified. Interestingly, despite the absence of the whole XIP subfamily and the NIP8 group, one *NIP2* and one *SIP3* were identified in this species. In *A. lyrata*, *A. halleri*, and saltwater cress, a number of 35, 36, and 35 *AQP* family genes were identified, respectively (Table S1). Despite their close relationships, compared with Arabidopsis, no *AtNIP1;1* ortholog was found in saltwater cress, the *AtPIP2;8* ortholog became a pseudogene, and the NIP4 group expanded via tandem duplication (Appendix A). Interestingly, in saltwater cress and *A. halleri*, two orthologs were identified for *AtTIP2;1*, i.e., *EsTIP2;1*, *EsTIP2;2*, *AhTIP2;1*, and *AhTIP2;2*, which were characterized as WGD repeats (Appendix A). Species-specific distribution of gene numbers in five subfamilies is summarized in Table 3. Generally, *AQP* family numbers are positively related to the total gene amounts in a genome but not always to the genome size, especially those that possess a high proportion of repetitive sequences such as rubber tree and *B. sinensis* (Figure 3). Arabidopsis was shown to possess the maximum density of *AQP* genes, rubber tree and *B. sinensis* harbor the minimum, and the density in papaya is comparative to *A. coerulea* and cassava (Table 3). Significantly, compared with *A. coerulea*, the PIP subfamily has highly expanded during the radiation of core eudicots, whereas the XIP subfamily is absent from spider flower and Brassicaceae plants (Table 3 and Appendix A), implying that its lineage-specific loss occurred sometime before the Brassicaceae–Cleomaceae divergence.

To infer species/lineage-specific evolution, the BRH-based sequence comparison was further used to identify OGs. As shown in Appendix A, a total of 61 OGs were obtained, and each evolutionary group possessed one to 14 OGs. A total of 29 *AcAQP* genes belonged to 21 OGs, eight of which had expanded via WGD (TIP1a and NIP1a), tandem duplication (NIP4 and XIP2), proximal duplication (XIP2), and dispersed duplication (TIP4, TIP6, and XIP2). Moreover, TIP1a/-2, TIP1d/-3, NIP1a/-2, NIP5/-6, and SIP1/-2 were characterized as transposed repeats, whereas TIP2/-4/-5/-6 and TIP1a/-d were characterized as dispersed repeats. This means that the early eudicot ancestor contained at least 21 members, i.e., two PIP1s, two PIP2s, two TIP1s, one TIP2, one TIP3, one TIP4, one TIP5, one TIP6, one NIP1, one NIP2, one NIP3, one NIP4, one NIP5, one NIP6, one NIP7, one XIP2, one SIP1, and one SIP2. During later evolution, significant expansion of several groups was observed in core eudicots, likely contributed by the γ WGD, i.e., PIP1, PIP2, TIP1, NIP4/-8, and XIP1/-2/-3. Except for NIP3a that is also found in Malpighiales plants, papaya shares all of the 20 other OGs identified in *A. coerulea* (Appendix A). Moreover, genes belonging to 12 out of these 20 OGs were shown to be located in syntenic blocks, exhibiting one-vs.-one, one-vs.-two, one-vs.-three, and two-vs.-one (Figure 4A), implying a conserved evolution of these genes even after the γ WGD. As for other genes, species-specific transposition or chromosomal rearrangement could be speculated. By contrast, more genes are located in syntenic blocks of Brassicales species, reflecting a relatively short time of evolution. Significantly, the majority of orthologs identified between papaya and horseradish are located in syntenic regions, only excluding *CpTIP2;1*/*MoTIP2;1*, *CpTIP3;1*/*MoTIP3;1*, *CpXIP2;1*/*MoXIP2;2*, and *CpSIP2;1*/*MoSIP2;1* (Figure 4B), though the horseradish genome is fragmented in 33,332 Scfs. It is worth noting that *MoXIP2;2* was a tandem repeat of *MoXIP2;1*, whereas *CpTIP2;1*, *MoTIP2;1*, *CpTIP3;1*, *MoTIP3;1*, *CpSIP2;1*, and *MoSIP2;1* were characterized as transposed repeats of *CpTIP1;1*, *MoTIP1;1*, *CpTIP1;2*, *MoTIP1;1*, *CpSIP1;1*, and *MoSIP1;1*, respectively (Appendix A). Since *CpTIP2;1*/*BsTIP2;1*, *CpXIP1;1*/*BsXIP1;1*, and *CpXIP2;1*/*BsXIP2;1* are located in syntenic blocks (Figure 4C), possible transposition or chromosomal rearrangement may occur in other species tested. Compared with *A. coerulea* (4), papaya (13), and horseradish (11), more duplicate genes identified in *B. sinensis* (35), spider flower (16), and Arabidopsis (20) are located in syntenic blocks (Figure 4), reflecting the occurrence of one to three additional WGD after the γ WGD. Although the spider flower genome is fragmented in 12,249 Scfs and Brassicaceae plants experienced the independent At-α WGD, the majority of duplicate genes identified in spider flower and Arabidopsis were shown to be located in syntenic blocks (Appendix A). Duplicated genes that are not located in syntenic blocks are as follows: *AtPIP2;3*, a tandem repeat of *AtPIP2;2*; *AtPIP2;4*, a syntelog of *CpPIP2;2*; *AtTIP1;2* and *AtTIP1;3*, two transposed repeats of *AtTIP1;1*; *AtNIP4;2*, a tandem repeat of *AtNIP4;1*; *ThPIP1;4*, a syntelog of *CpPIP1;2*; *ThPIP1;5*, an ortholog of *CpPIP1;3*; *ThPIP2;2*, a WGD repeat of *ThPIP2;1*; *ThTIP1;2*, a transposed repeat of *ThTIP1;1*; *ThTIP1;3*, a syntelog of *CpTIP1;3*; *ThTIP3;2*, a dispersed repeat of *ThTIP3;1*; *ThNIP2;1*, a syntelog of *CpNIP2;1*; *ThSIP3;1*, an ortholog of *CpSIP3;1* (Figure 4 and Appendix A). The presence of SIP3 and NIP2 members in spider flower implies (Appendix A) that their loss is Brassicaceae-specific, sometime after the split with spider flower.

### 2.3. Structural and Functional Inference of CpAQPs

As shown in Appendix A, gene structures were analyzed on the basis of curated gene models, and an example of comparing *A. coerulea*, papaya, and Arabidopsis *AQP* genes is shown in Figure 5B. Results revealed that the exon–intron structure is usually conserved within an evolutionary group and even subfamily, but variable between different subfamilies. Generally, TIP, SIP, and XIP feature two introns, whereas PIP and NIP possess three and four introns, respectively. Among three groups present in the SIP subfamily, SIP3 is the sole group without an intron. Differing from XIP2 and XIP3 within the XIP subfamily, XIP1 features a single intron, though *RcXIP1;3* was shown to be intronless. Besides the whole NIP5 group, *AtNIP1;3* and *AtNIP1;4* within the NIP subfamily also contain three introns, which appear to be conserved in Brassicaceae plants (Appendix A). TIP6 within the TIP subfamily features two introns. However, both *AtTIP6;1* and *AtTIP6;2* possess a single intron, which is also found in other Brassicaceae plants (Appendix A). The intron numbers in TIP1 vary from zero to three: TIP1a features a single intron, though *AcTIP1;2* was shown to contain three introns; TIP1d features two introns. However, *AtTIP1;3* and its orthologs in Brassicaceae plants are intronless; TIP1c is also intronless (Figure 5B and Appendix A).

Structural and possible functional divergence was investigated based on conserved motifs as well as dual NPA motifs, the NPA spacing, and the ar/R filter. According to the MEME analysis, 25 identified motifs were shown to be conserved within five subfamilies but distinct between different subfamilies (Figure 5C), reflecting their early divergence and a long time of evolution. Among them, PIP2s within the PIP subfamily possess 10 motifs, i.e., Motifs 1–7, 9, 12, 14, and 17, whereas PIP1s harbor one more (i.e., Motif 17) that is located at their extended N-terminus. TIPs usually contain 10 motifs, i.e., Motifs 1–3, 8, 10, 11, 16, 19, 21, and 24, though Motif 24 is absent from AcTIP1;1, AtTIP1;2, and three evolutionary groups (i.e., TIP2, TIP6, and TIP5) within the TIP subfamily. NIPs usually contain nine motifs, i.e., Motifs 1–3, 5, 13, 15, 18, 22, and 23, though loss of certain motifs was frequently found in three evolutionary groups (i.e., NIP5, NIP6, and NIP7) within the NIP subfamily. By contrast, subfamilies XIP and SIP were shown to contain considerably fewer motifs, i.e., five and four, respectively. Significantly, only two motifs, i.e., Motifs 3 and 20, were identified for both AtSIP2;1 and CpSIP3;1. Motif 3, which is found in all AQPs, is located in TM6 as well as TM3 in most members of subfamilies PIP, TIP, NIP, and XIP. Another two widely present motifs, i.e., Motifs 1 and 2, usually appear in two copies like Motif 3. Motif 1 spans TM2-HB and TM5-HE, which include the NPA motif and the ar/R filter. It is worth noting that the second copy of Motif 1 was replaced by Motif 25 and 20 in XIPs and SIPs, respectively. By contrast, little is known about Motif 2, which is usually located in TM1 and TM4 in contrast to being highly variable in the SIP subfamily. In addition to Motif 17, six other motifs are also specific to PIPs, i.e., Motifs 4, 6, 7, 9, 12, and 14: Motif 4 spans TM4-LD, which is replaced by Motif 8 or Motif 15 in TIPs and NIPs, respectively; Motif 6 spans TM1-LA-TM2, which is replaced by Motif 16 or Motif 22 in TIPs and NIPs, respectively; Motif 7 spans TM3-LC, which is replaced by Motif 10 (including an H residue at the position corresponding to H^131^ identified in AtTIP2;1) or Motif 23 in TIPs and NIPs, respectively; Motif 9 is located in TM6, which is replaced by Motif 19 or Motif 13 in TIPs and NIPs, respectively; Motif 12 is located in TM1, which is replaced by Motif 24 in TIPs; Motif 14 is located in LB and includes a putative phosphorylation site at the position corresponding to S115 identified in SoPIP2;1, which is replaced by Motif 21 or Motif 18 in TIPs and NIPs, respectively. In addition to Motifs 8, 10, 16, 21, and 24, Motif 11 is also specific to TIPs, which is located in LE and replaced by Motif 5 in four other subfamilies (Figure 5C).

In contrast to typical dual NPA motifs, NPV, NPT, NPI, NPS, NPL, NPT, NPC, NPG, and HPA variants were also observed, which are widely present in subfamilies SIP and XIP as well as four NIP groups (i.e., NIP1, NIP5, NIP6, and NIP7). The NPA spacing varies from 108 AA to 135 AA, where XIP features a relatively long NPA spacing. The spacing of 108 AA, which was proposed to be essential for silicon permeability, is not only found in NIP2 but also in most members of NIP5, NIP6, and NIP7 (Figure 5D). However, a T residue at the position corresponding to T^109^ in OsNIP2;1 was only found in NIPs (Appendix A), though the usual ar/R filter G-S-G-R was placed by G-V-G-R in AcNIP2;1 (Figure 5D). Corresponding to their substrate specificity, different families were shown to possess distinct ar/R filters. While the F-H-T-R ar/R filter is highly conserved in PIPs, TIPs usually harbor the H-I-A-V/R filter, though N/S-V/F-G-C variants were also observed; NIPs usually have the W-V/I-A-R filter, though A/G/S/T-I/V/S-G/A/S-R variants were also observed; XIP and SIP possess the V/I-V/I/T-V/A-R/K and V/S/A/I-V/H/K/T/F-P/G-I/S/N/A filters, respectively (Figure 5D). As shown in Appendix A, two highly conserved C residues were also identified in LC and HE of two CpXIPs; putative phosphorylation sites corresponding to S^274^ in SoPIP2;1 and S^262^ in GmNOD26 were found in all five CpPIP2s and CpNIP1;1/-4;1/-8;1, respectively; an H residue at the position corresponding to H^131^ in AtTIP2;1, which was proven to be essential for NH_3_ permeability, was found in CpTIP2;1, CpTIP4;1, and CpTIP6;1 (Appendix A).

### 2.4. Expression Patterns of CpAQP Genes

To uncover the expression evolution of *CpAQP* genes, their expression profiles were investigated based on RNA-seq data representing four main tissues that include two typical stages of developmental fruit, i.e., root, leaf, sap, and flesh of young and mature fruits. As shown in Figure 6, despite the expression of most *CpAQP* genes, their transcript levels were highly diverse. The total transcripts of the whole family were most abundant in the flesh of young fruits (100%), followed by the root (73.8%), moderate in the sap (22.7%) and the leaf (21.7%), and relatively low in the flesh of mature fruits (8.5%). Regardless of the tissue tested in this study, the majority of transcripts were contributed by PIPs and TIPs, varying from 82.7% in the leaf to 98.9% in the sap (Appendix A). In the root, 89.2% transcripts were contributed by eight genes, i.e., *CpPIP1;1*, *CpPIP1;3*, *CpPIP2;2*, *CpPIP2;3*, *CpPIP2;4*, *CpTIP1;1*, *CpTIP1;2*, and *CpTIP6;1*, where *CpTIP1;1* and *CpPIP1;3* represent the first and second most expressed *AQP* genes in this tissue. In the leaf, 81.1% transcripts were contributed by four genes, i.e., *CpPIP1;1*, *CpPIP2;4*, *CpTIP1;1*, and *CpXIP2;1*, where *CpTIP1;1* and *CpPIP1;1* represent the first and second most expressed *AQP* genes in this tissue. In the sap, 93.6% transcripts were contributed by three genes, i.e., *CpPIP1;1*, *CpPIP2;3*, and *CpTIP2;1*, where *CpPIP2;3* and *CpTIP2;1* represent the first and second most expressed *AQP* genes in this tissue. In the flesh of young fruits, 83.4% transcripts were contributed by three genes, i.e., *CpPIP1;1*, *CpPIP2;4*, and *CpTIP1;1*, where *CpTIP1;1* and *CpPIP2;4* represent the first and second most expressed *AQP* genes in this tissue. In the flesh of mature fruits, 88.6% transcripts were contributed by five genes, i.e., *CpPIP1;1*, *CpPIP2;4*, *CpTIP1;1*, *CpTIP1;3*, and *CpSIP1;1*, where *CpTIP1;1* and *CpPIP2;4* represent the first and second most expressed *AQP* genes in this tissue. Compared with the flesh of young fruits, the transcripts of most *AQP* genes were markedly down-regulated in the flesh of mature fruits, whereas *CpTIP1;3*, *CpSIP1;1*, and *CpTIP6;1* transcripts were significantly up-regulated. According to their expression patterns over different tissues, 29 *CpAQP* genes were grouped into three main clusters: Cluster I included the six most expressed genes, i.e., *CpPIP1;1*, *CpPIP1;3*, *CpPIP2;3*, *CpPIP2;4*, *CpTIP1;1*, and *CpTIP2;1*; Cluster II was moderately expressed, and included three groups, where IIa was predominantly expressed in the root as well as the flesh of mature fruits, IIb was preferentially expressed in the root, and IIc was typically expressed in the root as well as the leaf and the flesh of young fruits; Cluster III was lowly expressed or tissue-specific, and included four groups, where IIIa was preferentially expressed in the leaf, IIIb was lowly expressed in most tissues, IIIc was rarely expressed in most tissues, and IIId was typically expressed in the root (Figure 6). Interestingly, distinct expression patterns were frequently observed for duplicate pairs identified in this study, where *CpPIP1;1*, *CpPIP1;3*, *CpPIP2;3*, *CpPIP2;4*, and *CpTIP1;1* had evolved into the predominant isoforms, implying their subfunctionalization. By contrast, both *CpNIP4;1* and *CpNIP8;1* were shown to be rarely expressed in all tissues examined in this study, implying their possible functions in specific tissues and/or stages.

## 3. Discussion

Gene duplication, a prevalent phenomenon across the tree of life, has long been recognized as a key contributor to the evolution of genes with new functions [36,37,38]. Gene duplicates can arise from WGD as well as tandem, proximal, transposed, dispersed, and segmental duplications [39]. WGD, also known as polyploidy, which multiplies the whole genome content, had been proven to play an important role in the diversification of seed plants, angiosperms, as well as core eudicots [36,40]. Brassicales, an economically important order of flowering plants, represents 2.2% of the total extant core eudicot diversity and has been overlooked as a promising system to investigate patterns of disjunct distributions and diversification rates [41]. In addition to the γ event shared by all core eudicots [40], multiple independent WGDs have been described in Brassicales, e.g., one recent WGD in *B. sinensis*, the Th-α event within Cleomaceae, the At-α event at the base of Brassicaceae, and the At-β event near at the base of the order [15,42,43,44,45]. Despite the occurrence of two recent WGDs, Arabidopsis harbors a relatively small genome size of approximately 119.7 Mb and serves as a popular model species for research in many aspects of plant biology [16]. Soon after the first version of the Arabidopsis genome was released in 2000 [45], a genome-wide analysis was conducted to provide a systematic nomenclature for plant *AQP* genes [46]. The evolutionary analysis assigned 35 *AtAQP* genes into four subfamilies, i.e., PIP, TIP, NIP, and SIP, which were further divided into two to seven groups, i.e., PIP1–2, TIP1–5, NIP1–7, and SIP1–2 [46]. Surprisingly, a novel but ancient subfamily named XIP was later identified in moss, spikemoss, and a high number of eudicots [26,27,28,29,30,31,33]. Interestingly, this special subfamily was also shown to be absent from other Brassicaceae plants [32], implying that its loss may be lineage-specific. However, whether it is present in other Brassicales families is largely unknown. The accessibility of several representative Brassicales genomes beyond Brassicaceae, i.e., papaya in Caricaceae, horseradish in Moringaceae, *B. sinensis* in Akaniaceae, and spider flower in Cleomaceae [14,43,44,47], provides a good opportunity to address this issue.

In the current study, genome-wide identification of *AQP* family genes was performed in papaya as well as seven representative plant species, i.e., horseradish, *B. sinensis*, spider flower, *A. lyrata*, *A. halleri*, saltwater cress, and *A. coerulea*. In accordance with no recent WGD occurring in papaya and horseradish [3,47], small amounts of 29 and 28 *AQP* genes were identified from these two species, occupying 0.13% and 0.14% of the total protein-coding genes, respectively. The family numbers are equal or comparative to 29 (0.10%) and 31 (0.11%) members, respectively, found in *A. coerulea* and physic nut, but less than 35–55 (0.11–0.16%) members present in Arabidopsis, *A. lyrata*, *A. halleri*, saltwater cress, castor bean, cassava, rubber tree, *B. sinensis*, and poplar [29,30,33,35]. Like papaya and horseradish, comparative genomics analyses showed that no additional WGD was detected in both physic nut and castor bean after the γ event [48,49]. By contrast, the last common ancestor of cassava and rubber tree experienced one recent WGD [30,33,50], whereas poplar was proven to experience one Salicaceae-specific p WGD [34]. Interestingly, the amounts of *AQP* genes were shown to be positively associated with the total gene numbers present in genomes that were mainly contributed by recent WGDs [30,34,44,50], though tandem duplication plays an important role in the family expansion in castor bean [35]. Despite the occurrence of one ancient tetraploidization event in *A. coerulea* [51,52], only two WGD repeats are retained due to a long time of evolution. For species without recent WGDs, i.e., papaya, horseradish, physic nut, and castor bean, six to eight WGD repeats were identified, though a higher *E*-value cutoff of 1 × 10^−10^ instead of 1 × 10^−20^ was adopted in this study, which facilitates the detection of microsynteny and ancient WGD repeats [53]. By contrast, higher numbers of 11–26 WGD repeats were identified in poplar, cassava, rubber tree, *B. sinensis*, spider flower, and Brassicaceae species, corresponding to the occurrence of one to three recent WGDs in these species after the γ event [15,34,43,44,50]. In Arabidopsis, seven, three, and two repeats were shown to arise from the At-α, At-β, and γ events, respectively, most of which are conserved in Brassicaceae species examined in this study, though *A. halleri* and saltwater cress have retained one more repeat from the At-α WGD (i.e., *AhTIP2;1*/-*2;2* and *EsTIP2;1*/-*2;2*) and the ortholog of *AtPIP2;8* in saltwater cress is under fractionation. As reported in other plant species [29,30,33,35], the divergence of duplicate repeats identified in papaya is more likely to be constrained by purifying selection, since their Ka/Ks ratios are below one. Notably, compared with papaya, species-specific gene loss was also observed in its close species horseradish, i.e., orthologs of *CpNIP8;1* and *CpSIP3;1*.

Despite a relatively small family number, *CpAQP* genes represent all five previously defined subfamilies (i.e., PIP, TIP, NIP, SIP, and XIP) in higher plants [26,29] or 29 out of 61 OGs identified in this study, supporting their high diversity. More than two *XIP* genes found in papaya, horseradish, and *B. sinensis* but none in spider flower indicate that the complete loss of the whole XIP subfamily in Arabidopsis is lineage-specific, occurring sometime after its split with papaya but before the Brassicaceae–Cleomaceae divergence. Moreover, based on the comparative analysis of 530 *AQP* genes identified in 14 representative species, a new nomenclature for subclassification was proposed in this study, which includes 19 groups, i.e., PIP1–2, TIP1–6, NIP1–8, and SIP1–3. In this nomenclature, previously described *AtNIP2;1* and *-3;1*, which were characterized as transposed and WGD repeats of *AtNIP1;2* and *-1;1*, respectively, were assigned into the NIP1 group, whereas the proposed NIP3 group was consistent with that as described in Malpighiales plants [29,30,33,35]. The updated NIP2 group was also in accordance with previous studies, which was characterized as a silicon transporter widely present in monocots and eudicots [24,29,30,33,35,54,55]. The presence of this group in papaya, horseradish, *B. sinensis,* as well as spider flower supports the conjecure that its loss in Arabidopsis is Brassicaceae-specific, occurring sometime after the split with Cleomaceae. The previously described *AtTIP2;2* and *-2;3* were renamed *AtTIP6;1* and *-6;2*, respectively, and belong to the novel but ancient TIP6 group having diverged before the split of basal and core eudicots. Two other novel groups, i.e., NIP8 and SIP3, are widely present in the species examined in this study. The presence of SIP3 in spider flower implies that its loss in Brassicaceae is lineage-specific, occurring sometime after the split with Cleomaceae, whereas the absence of NIP8 from spider flower indicates that its loss in Brassicaceae species occurred sometime after its split with Caricaceae but before its split with Cleomaceae. Interestingly, species-specific loss of these two groups was also found in horseradish and cassava, whereas NIP8 is also absent from physic nut [29,30].

The updated subclassification is not only supported by evolutionary relationships, but also by exon–intron structures and/or conserved motifs/residues. Subfamily PIP, which mainly functions in water transport at the cell membrane [19], includes only two evolutionary groups and features three introns and the F-H-T-R ar/R filter as observed in the pure water channel AqpZ [56]. Compared with PIP2s, PIP1s possess longer N-terminal but shorter C-terminal sequences, and include the group-specific Motif 17 at the extended N-terminus. By contrast, PIP2s feature one putative phosphorylation site at the extended C-terminus as observed in SoPIP2;1 [19]. Notably, both groups especially PIP2 have highly expanded in core eudicots via WGDs as well as tandem duplication, forming 10 and 14 OGs as identified in this study, respectively, which is in accordance with their importance in adaptation [17,18,20]. Compared with PIP, Subfamily TIP, which is also highly permeable to water at the vacuolar membrane [57], is more diverse, possessing six evolutionary groups/16 OGs and harboring the variable ar/R filter of H/N/S-I/V/F-A/G-R/V/C. A total of seven motifs were identified to be specific to TIPs, i.e., Motifs 8, 10, 11, 16, 19, 21, and 24, though Motif 24 was absent from TIP2, TIP6, TIP5, and two members of TIP1. Similar to TIP, Subfamily NIP is also highly diverse, including eight evolutionary groups and 13 OGs with the ar/R filter of W/A/T/G/S-V/I/S-A/G/S-R. Five motifs identified in this study were shown to be specific to NIPs, i.e., Motifs 13, 15, 18, 22, and 23, though Motifs 13 and 23 were absent from NIP5, NIP6, and NIP7. NIP5, which features the unusual NPS-NPV motifs, represents the unique group with three introns within the NIP subfamily. As for two other distinct subfamilies, Subfamily XIP features longer NPA spacing and two conserved C residues, whereas Subfamily SIP usually possesses a relatively short N-terminus, though both of them feature two introns and a few conserved motifs. Atypical NPL/T/S/C at the first NPA motif was found in all SIPs, which were shown to localize to the endoplasmic reticulum (ER) [58]. Among the three evolutionary groups present in Subfamily SIP, SIP3 differed from other groups with no introns, whereas SIP1 and SIP2 favored the NPT and NPL, respectively. Although SIPs possess the distinct and variable ar/R filters from PIPs and TIPs, SIP1 members have been proven to transport water [58,59]. By contrast, plasma membrane-localized XIPs, which are widely present in plants, moss, fungi, and protozoan species [31], have been reported to transport glycerol, urea, boric acid, and H_2_O_2_ but not water [60,61,62,63]. The complete loss of Subfamily XIP in monocots as well as two families within Brassicales, i.e., Brassicaceae and Cleomaceae, may be due to functional redundancy with other AQP subfamilies, e.g., NIP, which was proven to transport water, glycerol, urea, arsenite, selenite, boric acid, lactic acid, silicic acid, NH_3_, and H_2_O_2_ [54,64,65,66,67,68,69,70,71,72]. Unlike other subfamilies, the expansion of Subfamily XIP was mainly a result of tandem duplication, composed of three evolutionary groups with NPV/T/I and SPT/I/A/V variants at the first NPA motif. XIP1 differs from other groups with a single intron, implying its possible retrotransposition origin as described in Arabidopsis [46].

Orthologs, which evolved from a common ancestral gene via speciation, usually retain the same functions in the course of evolution in different species [73]. The characterization of OGs and the gene expression profiling conducted in this study allow us to infer putative roles of *AQP* genes in papaya. In agreement with previous studies [29,30,35,74], *AQP* transcripts in all five samples (i.e., leaf, root, sap, the flesh of young fruits, and the flesh of mature fruits) examined in this study were mainly contributed by PIPs and TIPs, which mediate the water transport at the plasma and vacuolar membranes, respectively [17,20,57]. Nevertheless, the expression patterns of different family members appear to be tissue-specific. For example, among five tissues/stages examined in this study, *CpAQP* transcripts were shown to be most abundant in the flesh of young fruits, corresponding to rapid cell enlargement as well as high water content in early stages of fruit development [75,76,77,78]. Moreover, in the flesh of young fruits, most transcripts were contributed by *CpTIP1;1*, *CpPIP2;4*, and *CpPIP1;1*, in contrast to the flesh of mature fruits by *CpTIP1;1*, *CpPIP2;4*, *CpTIP1;3*, and *CpPIP1;1*. In the leaf, a more important role of *CpPIP1;1* was observed, i.e., *CpTIP1;1*, *CpPIP1;1*, and *CpPIP2;4*, in contrast to the root by *CpTIP1;1*, *CpPIP1;3*, *CpTIP6;1*, and *CpPIP2;2*. By contrast, in the sap, most transcripts were mainly contributed by two members, i.e., *CpPIP2;3* and *CpTIP2;1*, which is highly different from other tissues. It is well known that leaves are photosynthetic organs that regulate water loss through transpiration; roots function in regulating water and nutrient uptake, and the phloem sap is responsible for the movement, distribution, and trafficking of water, nutrients, photoassimilates, and other macromolecules [17,79,80]. A more important role of TIP2s than of TIP1s was also described in the root of the rubber tree [30]. Whereas TIP1s transport water and urea [57,81], TIP2s and TIP6s facilitate the transport of water and NH_3_ [23,82]. Compared with PIPs and TIPs, members of three other subfamilies were less expressed, most of which belong to IIIa to IIIc identified in this study. Generally, NIPs and XIPs transport small solutes rather than water [60,61,62,63,64,65,66,67,68,69,70,71,72]. Interestingly, a recent study revealed a link between *AQP* expression and the texture of papaya fruit under different cultivation conditions. Compared with open field conditions, mesocarp cells and intercellular spaces of papaya fruit were larger when cultivated in raised beds, which were shown to correlate with higher expressions of *CpTIP2;1*, *CpTIP4;1*, *CpSIP1;1* and *CpPIP1;3*. Moreover, expressions of *CpTIP2;1*/*-4;1* and *CpSIP1;1/CpPIP2;5* correlated with the fruit crispness under open field and raised bed conditions, respectively [83].

## 4. Materials and Methods

### 4.1. Datasets and Sequence Retrieval

*AQP* genes that were previously described in Arabidopsis, castor bean, physic nut, cassava, rubber tree, and poplar were acquired according to related literature [29,30,31,33,35,46], and detailed information is shown in Appendix A. Genome sequences of papaya Sunset and *B. sinensis* were downloaded from the Genome Warehouse (GWH) database in BIG Data Center (https://ngdc.cncb.ac.cn/search/?dbId=gwh&q=GWHBFSD00000000 or https://ngdc.cncb.ac.cn/search/?dbId=gwh&q=GWHBDNO00000000, accessed on 31 June 2023), whereas genome and transcriptome data of papaya SunUp (ASGPBv0.4), horseradish (v1), spider flower (ASM46358v1), saltwater cress (v1.0), *A. halleri* (v2.1), *A. lyrata* (v2.1), and *A. coerulea* (v3.1) were accessed from NCBI (http://www.ncbi.nlm.nih.gov/, accessed on 31 June 2023) and Phytozome v13 (https://phytozome-next.jgi.doe.gov/, accessed on 31 June 2023).

### 4.2. Identification and Manual Curation of AQP Family Genes

A homology search was performed using published AQP proteins as queries, where the *E*-value of tBLASTn was set to 1 × 10^−5^. Positive genomic sequences were predicted as previously described [35], and all gene models were further validated with mRNA when available. A homology search for nucleotides or expressed sequence tags (ESTs) was conducted using BLASTn, and read alignment of RNA sequencing (RNA-seq) data was carried out using Bowtie 2 [84]. The presence of the conserved MIP domain (Pfam accession number PF00230) in deduced peptides was confirmed using the Pfam search (v35.0, https://pfam.xfam.org/, accessed on 31 June 2023).

### 4.3. Synteny Analysis and Gene Evolution Patterns

Homolog pairs within and between species were identified using the all-to-all BLASTP method with the *E*-value of 1 × 10^−10^, whereas syntenic blocks (BLAST hits ≥ 5) and gene collinearity were inferred using MCScanX as previously described [85]. WGD repeats were defined when homolog pairs are located within syntenic blocks of duplicated Chrs/Scfs, while tandem repeats were considered when two paralogs were consecutive in a genome. Transposed, proximal, and dispersed repeats were identified using the DupGen_finder pipeline as described before [39]. To uncover the evolutionary rate of duplicate pairs, Ka (nonsynonymous substitution rate) and Ks (synonymous substitution rate) were calculated by codeml in the PAML package [86]. Orthologs across different species were identified using the BRH (best reciprocal hit) method [87] as well as information from synteny analysis, and orthogroups (OGs) were assigned only when they were present in at least two species tested.

### 4.4. Sequence Alignment, Evolutionary Analysis, and Classification

Multiple sequence alignment of full-length AQP proteins was performed using MUSCLE [88]. Unrooted trees were constructed using MEGA 6.0 [89] with the parameters as follows: maximum likelihood method, bootstrap of 1000 replicates, Jones–Taylor–Thornton (JTT) model, uniform rates, complete deletion of gaps, nearest-neighbor interchange (NNI), and making initial tree automatically (Default-NJ/BioNJ). Except for three novel groups identified in this study, the classification of AQPs into subfamilies and groups was performed as described before [35].

### 4.5. Sequence and Structural Features

Exon–intron structures were displayed using GSDS 2.0 (http://gsds.gao-lab.org/, accessed on 31 June 2023). Protein features such as theoretical molecular weight (MW), isoelectric point (pI), grand average of hydropathicity (GRAVY), aliphatic index (AI), and instability index (II) were calculated using ProtParam (http://web.expasy.org/protparam/, accessed on 31 June 2023). Functional prediction was performed based on analysis of dual NPA motifs, the NPA spacing, and the ar/R selectivity filter from alignments with the structure resolved spinach (*Spinacia oleracea*) PIP2;1, AtTIP2;1, and OsNIP2;1 as well as functionally characterized AQPs [18,22,24]. A homology model was conducted using SWISS-MODEL (https://swissmodel.expasy.org/interactive, accessed on 31 June 2023) and AlphaFold 2 (https://colab.research.google.com/github/deepmind/alphafold/blob/main/notebooks/AlphaFold.ipynb, accessed on 31 June 2023). Additionally, conserved motifs in Ac/Cp/AtAQP proteins were analyzed using MEME (v5.5.0) [90], and the optimized parameters were as follows: any number of repetitions; maximum number of motifs, 25; and, the optimum width of each motif, between 6 and 50 residues.

### 4.6. Gene Expression Analysis

Expression profiles of *CpAQP* genes were investigated based on Illumina RNA-seq samples as shown in Appendix A. Leaves, roots, and phloem sap were collected from three-month-old plants of the Maradol Roja variety grown under greenhouse conditions. Immature and mature flesh that were white or yellow in color were obtained from fruits of green (young) and color break (mature) stages, respectively. Quality control of raw RNA-seq reads was performed using Trimmomatic [91], and read mapping was carried out using Bowtie 2 [84]. The FPKM (fragments per kilobase of exon per million fragments mapped) method [92] was adopted for expression annotation, and RSEM (v1.2.27) [93] was used to determine differentially expressed genes. Unless specifically stated, the tools in this study were used with default parameters.

## 5. Conclusions

To our knowledge, this is the first genome-wide analysis of the *AQP* gene family in papaya. A relatively small number of 29 members were shown to be highly diverse, representing all five subfamilies, 22 evolutionary groups, and 29 out of 61 OGs identified in this study. A further comprehensive comparison with *AQP* genes identified from 13 other representative plant species (totaling 530 *AQP* genes) provides insights into lineage-specific family evolution in Brassicales, including lineage-specific loss of the XIP subfamily and several evolutionary groups such as NIP2, NIP3, NIP8, and SIP3. Moreover, characterization of OGs, the ar/R filter, and gene expression profiles facilitates further functional studies of *AQP* genes in papaya and other species.

## Figures and Tables

**Figure 1 plants-12-03847-f001:**
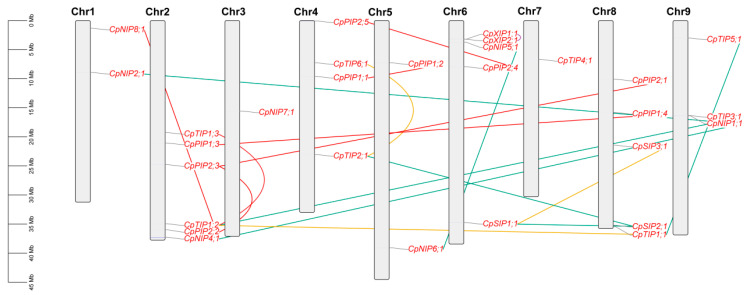
Chromosomal locations and duplication events of 29 *CpAQP* genes. Serial numbers are indicated at the top of each chromosome, and the scale is in Mb. Duplicate pairs identified in this study are connected using lines in different colors, i.e., tandem (purple), transposed (blue), dispersed (gold), and WGD (red). (AQP: aquaporin; Chr: chromosome; Cp: *C. papaya*; Mb: megabase; NIP: NOD26-like intrinsic protein; PIP: plasma intrinsic membrane protein; SIP: small basic intrinsic protein; TIP: tonoplast intrinsic protein; XIP: X intrinsic protein).

**Figure 2 plants-12-03847-f002:**
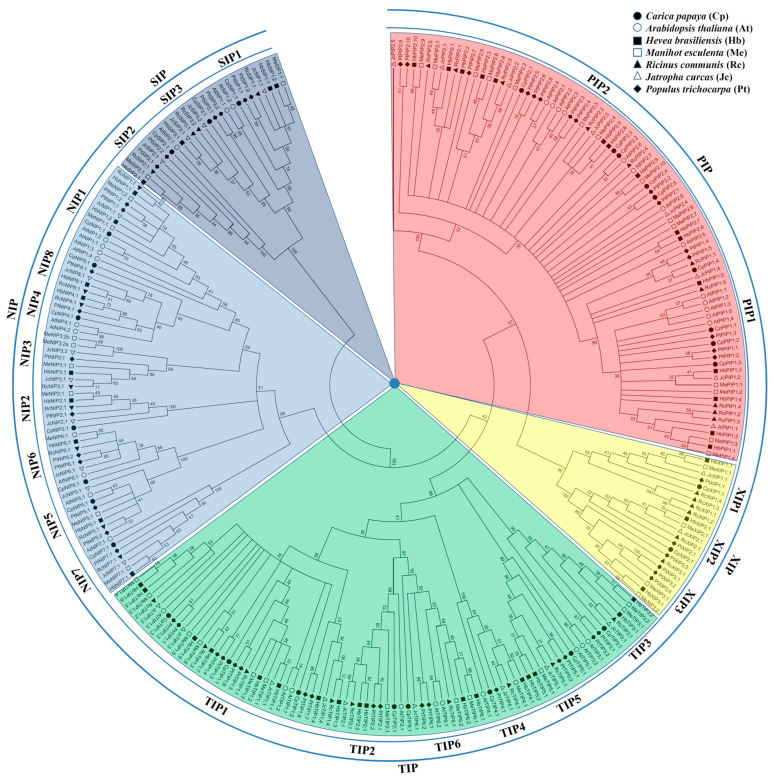
Evolutionary analysis of AQPs present in *C. papaya*, *A. thaliana*, *R. communis*, *J. curcas*, *M. esculenta*, *H. brasiliensis*, and *P. trichocarpa*. Sequence alignment was performed using MUSCLE, and the unrooted evolutionary tree was constructed using the bootstrap maximum likelihood tree (1000 replicates) method of MEGA 6.0. The distance scale denotes the number of amino acid substitutions per site, and the name of each subfamily/group is indicated next to the corresponding cluster. (AQP: aquaporin; At: *A. thaliana*; Cp: *C. papaya*; Hb: *H. brasiliensis*; Jc: *J. curcas*; Me: *M. esculenta*; NIP: NOD26-like intrinsic protein; PIP: plasma intrinsic membrane protein; Pt: *P. trichocarpa*; Rc: *R. communis*; SIP: small basic intrinsic protein; TIP: tonoplast intrinsic protein; XIP: X intrinsic protein).

**Figure 3 plants-12-03847-f003:**
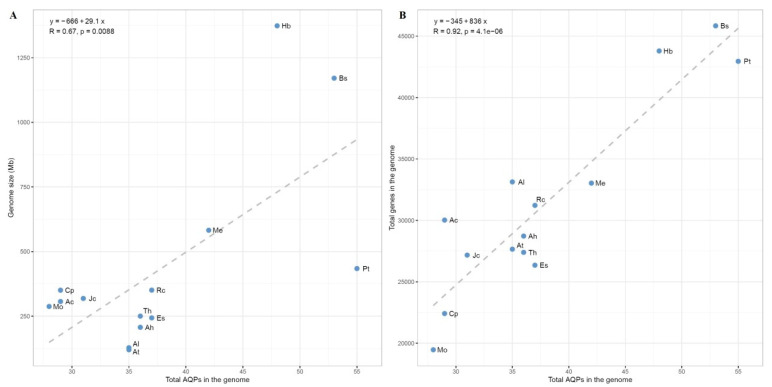
Relationships between the number of *AQP* genes, the genome size, and the total number of predicted protein-coding genes in genomes. Total *AQP* genes identified in *C. papaya*, *M. oleifera*, *B. sinensis*, *T. hassleriana*, *E. salsugineum*, *R. communis*, *J. curcas*, *M. esculenta*, *H. brasiliensis*, *P. trichocarpa*, and *A. coerulea* was plotted against (**A**) the genome size and (**B**) the total number of predicted genes. (Ac: *A. coerulea*; Ah: *A. halleri*; Al: *A. lyrata*; AQP: aquaporin; At: *A. thaliana*; Bs: *B. sinensis*; Cp: *C. papaya*; Es: *E. salsugineum*; Hb: *H. brasiliensis*; Jc: *J. curcas*; Me: *M. esculenta*; Mo: *M. oleifera*; Pt: *P. trichocarpa*; Rc: *R. communis*; Th: *T. hassleriana*).

**Figure 4 plants-12-03847-f004:**
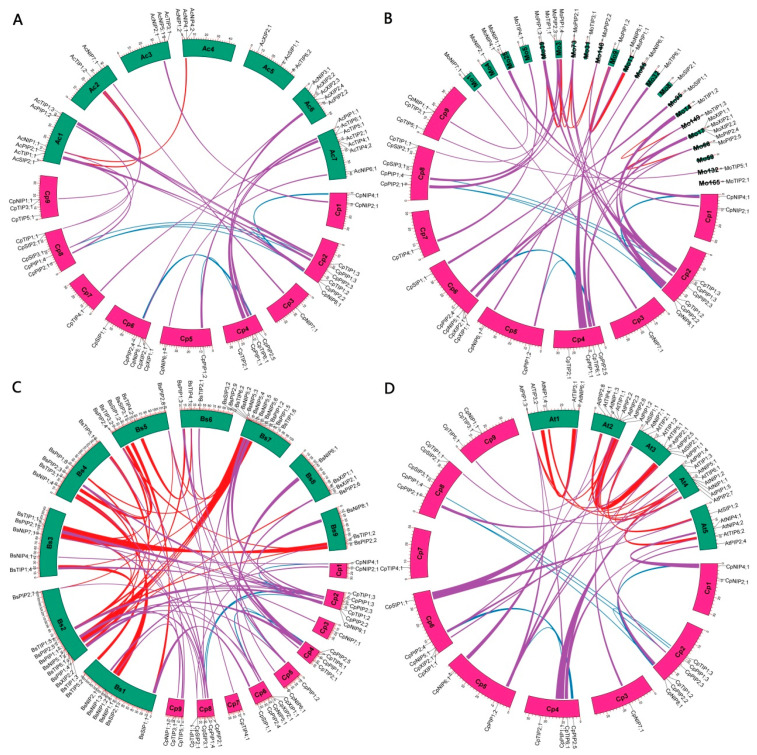
Synteny analyses within and between *C. papaya* and other species. (**A**) *C. papaya* and *A. coerulea*. (**B**) *C. papaya* and *M. oleifera*. (**C**) *C. papaya* and *B. sinensis*. (**D**) *C. papaya* and *A. thaliana*. Lines shown in different colors, i.e., *C. papaya* (blue), *A. coerulea*/*M. oleifera*/*B. sinensis*/*A. thaliana* (red), and papaya with *A. coerulea*/*M. oleifera*/*B. sinensis***/***A. thaliana* (purple), connect AQP-encoding syntenic blocks, which were inferred using MCScanX (*E*-value ≤ 1 × 10^−10^; BLAST hits ≥ 5). (Ac: *A. coerulea*; At: *A. thaliana*; AQP: aquaporin; Bs: *B. sinensis*; Chr: chromosome; Cp: *C. papaya*; Mo: *M. oleifera*; NIP: NOD26-like intrinsic protein; PIP: plasma intrinsic membrane protein; Scf: scaffold; SIP: small basic intrinsic protein; TIP: tonoplast intrinsic protein; XIP: X intrinsic protein).

**Figure 5 plants-12-03847-f005:**
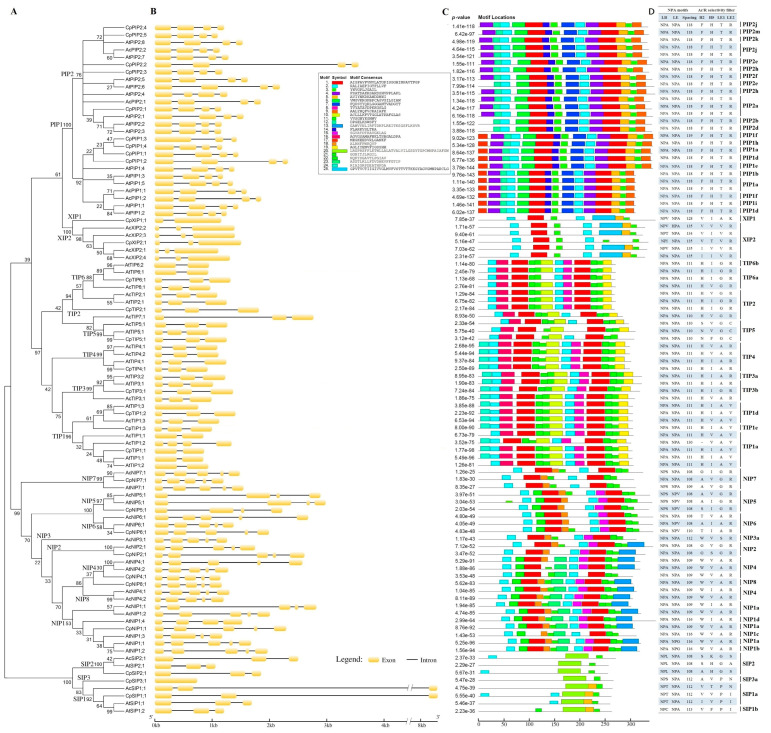
Sequence and structural features of *C. papaya*, *A. thaliana*, and *A. coerulea* AQPs. (**A**) Shown is the unrooted evolutionary tree resulting from full-length AQPs with MEGA 6.0 (MUSCLE, maximum likelihood method, and bootstrap of 1000 replicates). The distance scale denotes the number of amino acid substitutions per site, and the name of each group is indicated next to the corresponding cluster. (**B**) Shown are the exon–intron structures displayed using GSDS 2.0. (**C**) Shown is the distribution of conserved motifs among AQPs, where different motifs are represented by different color blocks as indicated at the left of the figure and the same color block in different proteins indicates a certain motif. (**D**) Shown are the dual NPA motifs, the NPA spacing, and the ar/R selectivity filter identified in this study. (Ac: *A. coerulea*; AQP: aquaporin; ar/R: aromatic/arginine; At: *A. thaliana*; Cp: *C. papaya*; NIP: NOD26-like intrinsic protein; NPA: Asn-Pro-Ala; PIP: plasma intrinsic membrane protein; SIP: small basic intrinsic protein; TIP: tonoplast intrinsic protein; XIP: X intrinsic protein).

**Figure 6 plants-12-03847-f006:**
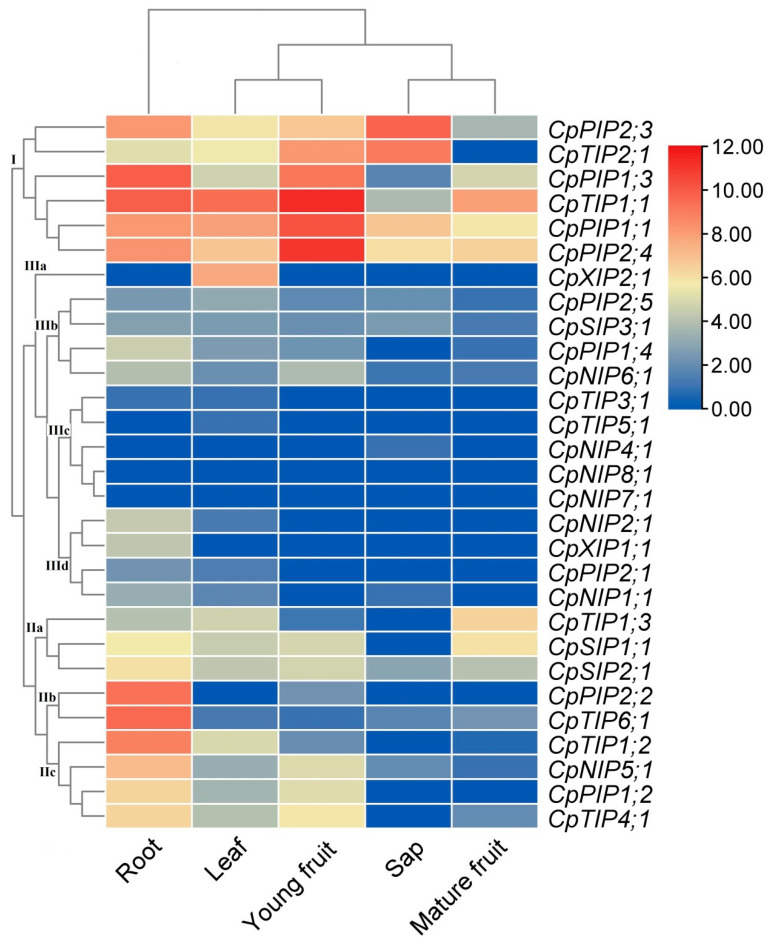
Expression patterns of *CpAQP* genes in various tissues and different stages of developmental fruit. Color scale represents FPKM normalized log_2_ transformed counts where blue indicates low expression and red indicates high expression. (AQP: aquaporin; Cp: *C. papaya*; FPKM: fragments per kilobase of exon per million fragments mapped; NIP: NOD26-like intrinsic protein; PIP: plasma intrinsic membrane protein; SIP: small basic intrinsic protein; TIP: tonoplast intrinsic protein; XIP: X intrinsic protein).

**Table 1 plants-12-03847-t001:** *AQP* family genes identified in *C. papaya*. (AA: amino acid; AI: aliphatic index; AQP: aquaporin; Cp: *C. papaya*; EST: expressed sequence tag; GRAVY: grand average of hydropathicity; II: instability index; kDa: kilodalton; MW: molecular weight; NIP: NOD26-like intrinsic protein; pI: isoelectric point; PIP: plasma intrinsic membrane protein; SIP: small basic intrinsic protein; TIP: tonoplast intrinsic protein; XIP: X intrinsic protein).

Gene Name	Position		EST Hits	AA	MW (kDa)	pI	GRAVY	AI	II
	Sunset	ASGPBv0.4							
*CpPIP1;1*	Chr4:9607168..9608940(+)	Supercontig74:37848..39306(−)	21	286	30.83	8.86	0.327	92.83	28.19
*CpPIP1;2*	Chr5:7251541..7253955(+)	Supercontig11:292060..294383(+)	7	286	30.64	8.84	0.392	97.62	29.04
*CpPIP1;3*	Chr2:21050877..21052615(+)	Supercontig19:1476407..1477835(−)	20	289	30.79	8.60	0.367	96.92	30.30
*CpPIP1;4*	Chr8:15800164..15801507(−)	Supercontig163:333798..334973(−)	3	287	30.80	9.00	0.426	99.65	32.10
*CpPIP2;1*	Chr8:10070161..10071393(−)	Supercontig102:347371..348540(−)	0	285	30.45	8.54	0.396	97.58	34.68
*CpPIP2;2*	Chr2:24775315..24779320(+)	Supercontig53:531406..534958(−)	63	281	29.99	9.03	0.554	98.65	32.45
*CpPIP2;3*	Chr2:35976240..35976749(−)	Supercontig216:63055..63993(+)	3	286	30.57	7.70	0.495	102.38	30.57
*CpPIP2;4*	Chr6:7915928..7916372(−)	Supercontig20:1055211..1056420(+)	19	279	29.70	9.25	0.508	103.23	33.37
*CpPIP2;5*	Chr4:4319017..4320116(+)	Supercontig6:2241234..2242332(+)	0	280	29.78	8.33	0.474	99.68	37.20
*CpTIP1;1*	Chr8:35459118..35459861(−)	Supercontig92:760280.761359(+)	16	251	25.78	5.78	0.783	105.50	26.77
*CpTIP1;2*	Chr2:34940054..34941644(−)	Supercontig190:276233..277641(−)	35	252	25.87	5.38	0.874	109.68	29.75
*CpTIP1;3*	Chr2:19257538..19258799(+)	Supercontig1:4585084..4586310(+)	31	252	26.20	5.78	0.754	101.51	25.12
*CpTIP2;1*	Chr4:23042560..23044618(−)	Supercontig109:530931..531183(+)	11	248	25.17	6.00	1.063	119.27	33.04
*CpTIP6;1*	Chr4:7255686..7257121(+)	Supercontig2391:4680..6000(−)	18	249	25.09	4.83	0.975	117.15	27.65
*CpTIP3;1*	Chr9:16333844..16335209(−)	Supercontig829:83522..84888(−)	0	263	27.90	6.97	0.593	113.80	38.64
*CpTIP4;1*	Chr7:6680977..6682047(+)	Supercontig7:2553814..2554736(+)	22	247	26.05	6.27	0.774	115.63	23.77
*CpTIP5;1*	Chr9:3010290..3011536(+)	Supercontig3:2267325..2268565(+)	0	259	26.76	8.71	0.661	98.69	37.81
*CpNIP1;1*	Chr9:16371472..16374264(+)	Supercontig124:278931..281224(+)	0	285	30.55	8.77	0.489	100.95	33.65
*CpNIP2;1*	Chr1:8925416..8928438(−)	Supercontig140:711835..714446(+)	0	292	31.05	8.94	0.424	104.59	34.28
*CpNIP4;1*	Chr1:1326996..1328142(−)	Contig28186:8449..9595(−)	0	270	28.36	5.35	0.589	102.22	27.44
*CpNIP8;1*	Chr2:37253990..37278417(−)	Supercontig65:1214676..1215502(−)	0	270	29.38	8.43	0.656	113.63	47.11
*CpNIP5;1*	Chr6:3733586..3736229(−)	Supercontig37:1035251..1037475(+)	0	298	30.87	8.78	0.517	101.24	33.19
*CpNIP6;1*	Chr5:39067672..39069664(−)	Supercontig10:1520284..1522276(+)	0	310	32.30	8.71	0.424	97.65	37.00
*CpNIP7;1*	Chr3:15553029..15554003(−)	Supercontig147:143267..144241(+)	1	272	29.16	7.63	0.779	109.01	31.57
*CpXIP1;1*	Chr6:3206824..3208026(−)	Supercontig37:1572478..1573637(+)	0	322	34.73	8.05	0.576	100.87	34.70
*CpXIP2;1*	Chr6:3208717..3210258(−)	Supercontig37:1570247..1571752(+)	1	309	32.86	9.32	0.708	117.02	26.97
*CpSIP1;1*	Chr6:34723643..34729027(−)	Supercontig676:90..389(+)	14	242	25.82	9.52	0.856	112.15	27.00
*CpSIP3;1*	Chr8:21431405..21432342(−)	Contig33833:1599..2339(−)	0	246	26.25	10.42	0.652	114.51	25.77
*CpSIP2;1*	Chr8:35197612..35199896(+)	Supercontig92:495190..497045(+)	0	236	25.70	9.66	0.672	114.45	18.00

**Table 2 plants-12-03847-t002:** *AQP* duplicates identified in *C. papaya*. Ks and Ka were calculated using PAML. (AQP: aquaporin; Cp: *C. papaya*; Ka: nonsynonymous substitution rate; Ks: synonymous substitution rate; NIP: NOD26-like intrinsic protein; PIP: plasma intrinsic membrane protein; SIP: small basic intrinsic protein; TIP: tonoplast intrinsic protein; WGD: whole-genome duplication; XIP: X intrinsic protein).

Duplicate 1	Duplicate 2	Ks	Ka	Ka/Ks	Mode
*CpPIP1;2*	*CpPIP1;1*	1.1851	0.0547	0.0461	WGD
*CpPIP1;4*	*CpPIP1;3*	1.5672	0.0948	0.0605	WGD
*CpPIP2;2*	*CpPIP2;1*	2.5002	0.0984	0.0394	WGD
*CpPIP2;3*	*CpPIP2;2*	1.8078	0.0730	0.0404	WGD
*CpPIP2;5*	*CpPIP2;4*	1.4392	0.1624	0.1128	WGD
*CpTIP1;3*	*CpTIP1;2*	2.5554	0.1384	0.0542	WGD
*CpNIP8;1*	*CpNIP4;1*	3.2959	0.3236	0.0982	WGD
*CpXIP1;1*	*CpXIP2;1*	3.8857	0.5667	0.1458	Tandem
*CpTIP6;1*	*CpTIP2;1*	1.8799	0.1972	0.1049	Dispersed
*CpTIP2;1*	*CpTIP1;1*	40.7578	0.3118	0.0077	Transposed
*CpTIP5;1*	*CpTIP1;1*	41.3254	0.5372	0.0130	Transposed
*CpTIP3;1*	*CpTIP1;2*	22.9755	0.3046	0.0133	Transposed
*CpNIP2;1*	*CpNIP1;1*	44.4070	0.5210	0.0117	Transposed
*CpNIP4;1*	*CpNIP1;1*	3.5397	0.4271	0.1206	Transposed
*CpNIP5;1*	*CpNIP6;1*	40.4672	0.2635	0.0065	Transposed
*CpSIP2;1*	*CpSIP1;1*	42.6702	0.8106	0.0190	Transposed
*CpTIP1;1*	*CpTIP1;2*	3.2492	0.1272	0.0391	Dispersed
*CpSIP1;1*	*CpSIP3;1*	3.6631	0.3916	0.1069	Dispersed

**Table 3 plants-12-03847-t003:** Species-specific distribution of AQP members in five subfamilies. (Mb: megabase; NIP: NOD26-like intrinsic protein; PIP: plasma intrinsic membrane protein; SIP: small basic intrinsic protein; TIP: tonoplast intrinsic protein; XIP: X intrinsic protein).

Family	Species	Genome Size (Mb)	Protein-Coding Genes	Gene Numbers						Genes/Mb	
				PIP	TIP	NIP	SIP	XIP	Total	Total	AQP
Ranunculaceae	*A. coerulea*	306.5	30,023	4	10	9	4	2	29	97.95	0.09
Salicaceae	*P. trichocarpa*	434.1	42,950	15	17	11	6	6	55	98.94	0.13
Euphorbiaceae	*R. communis*	350.6	31,221	10	9	8	4	6	37	89.05	0.11
Euphorbiaceae	*J. curcas*	318.4	27,172	8	9	8	4	2	31	85.34	0.10
Euphorbiaceae	*M. esculenta*	582.3	33,033	14	13	9	4	2	42	56.73	0.07
Euphorbiaceae	*H. brasiliensis*	1373.5	43,792	15	17	9	3	4	48	31.88	0.03
Akaniaceae	*B. sinensis*	1170.9	45,839	15	13	15	3	5	53	39.15	0.05
Caricaceae	*C. papaya*	350.3	22,416	9	8	7	2	3	29	63.99	0.08
Moringaceae	*M. oleifera*	287.4	19,465	9	8	6	3	2	28	67.73	0.10
Cleomaceae	*T. hassleriana*	249.9	27,396	13	10	8	4	0	36	109.63	0.14
Brassicaceae	*E. salsugineum*	243.1	26,351	12	11	9	3	0	35	108.40	0.14
Brassicaceae	*A. halleri*	195.6	28,722	13	11	9	3	0	36	146.84	0.18
Brassicaceae	*A. lyrata*	206.6	33,132	13	10	9	3	0	35	160.37	0.17
Brassicaceae	*A. thaliana*	119.7	27,655	13	10	9	3	0	35	231.04	0.29

## Data Availability

SRA accession numbers of transcriptome data used in this study are shown in Appendix A.

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
