# Peer review of "Analysis of Carica papaya Informs Lineage-Specific Evolution of the Aquaporin (AQP) Family in Brassicales"

_plants, 2023, doi:10.3390/plants12223847_

Round 1
Reviewer 1 Report
Comments and Suggestions for Authors
Review of Genomic analysis of Carica papaya aquaporin genes provides insights into lineage-specific family evolution in Brassicales by Zou et al. for Plants
I am recommending this manuscript for minor revisions. I appreciate much about this manuscript, especially its focus on gene family evolution within a lineage rather than other papers, which can take a kitchen-sink style approach to genome analysis. Most analyses seem sound, as do their interpretations. I do, however, have three sizable reservations about the title, description of phylogenetic analyses, and interpretation of Ka/Ks values that need to be addressed before this paper is published.
It is not clear from the title if the paper is discussing a taxonomic family of plants or a gene family. Much of the title is vague and superfluous. I suggest this alternative title:
“Analysis of Carica papaya informs lineage-specific aquaporin gene family evolution in Brassicales”
Line 153 – What molecular evolution model was used for maximum likelihood analyses in MEGA? How was the choice of model made? Was a model testing program used? If so, what program was used and what settings? Was AIC or BIC used for model testing? Including these details is essential for replicability for maximum likelihood phylogenetic analyses—many settings and options are chosen when performing these analyses. There are many choices made in initiating a phylogenetic analysis in MEGA and these choices, and their rationales, need to be included in the methods.
Line 267-269 – Just because these gene duplicates are under purifying selection, it does not mean that their divergence is driven by purifying selection—indeed they would be more diverged were it not for purifying selection. Purifying selection constrains divergence; it does not drive it. It would be more appropriate to say that their evolution was constrained by purifying selection. Indeed, high Ks values suggest that their divergence was driven by a high intrinsic mutation rate (and perhaps facilitated by drift) that was mitigated to some degree by purifying selection, as evidenced by the low Ka.
At the very least, this sentence:
“Given the Ka/Ks ratios are all below one (from 0.0258 to 0.3751) (Table 2), their divergence is more likely to be driven by purifying selection.”
should be changed to something like this:
“Given the Ka/Ks ratios are all below one (from 0.0258 to 0.3751) (Table 2), their divergence is likely to be constrained by purifying selection.”
and the authors can elaborate further if they feel so compelled (if they do so, I hope they keep my comments in mind).
I look forward to seeing this paper published after addressing my small but important reservations.

Author Response
Response to Reviewer #1:
Comment: Review of Genomic analysis of Carica papaya aquaporin genes provides insights into lineage-specific family evolution in Brassicales by Zou et al. for Plants
I am recommending this manuscript for minor revisions. I appreciate much about this manuscript, especially its focus on gene family evolution within a lineage rather than other papers, which can take a kitchen-sink style approach to genome analysis. Most analyses seem sound, as do their interpretations. I do, however, have three sizable reservations about the title, description of phylogenetic analyses, and interpretation of Ka/Ks values that need to be addressed before this paper is published.
Response: Thank you for your positive consideration for our manuscript. This study aimed to fill the gap whether XIPs are present in other Brassicales families beyond Brassicaceae and lineage-specific evolution patterns of the whole gene family in Brassicales.
Comment: It is not clear from the title if the paper is discussing a taxonomic family of plants or a gene family. Much of the title is vague and superfluous. I suggest this alternative title:
“Analysis of Carica papaya informs lineage-specific aquaporin gene family evolution in Brassicales”
Response: As suggested by the reviewer, the title has been changed to “Analysis of Carica papaya informs lineage-specific evolution of the aquaporin (AQP) family in Brassicales” in the revised version of manuscript.
Comment: Line 153 – What molecular evolution model was used for maximum likelihood analyses in MEGA? How was the choice of model made? Was a model testing program used? If so, what program was used and what settings? Was AIC or BIC used for model testing? Including these details is essential for replicability for maximum likelihood phylogenetic analyses—many settings and options are chosen when performing these analyses. There are many choices made in initiating a phylogenetic analysis in MEGA and these choices, and their rationales, need to be included in the methods.
Response: As suggested by the reviewer, detailed parameters have been supplemented as “bootstrap of 1,000 replicates, Jones-Taylor-Thornton (JTT) model, uniform rates, complete deletion of gaps, Nearest-Neighbor-Interchange (NNI), and make initial tree automatically (Default-NJ/BioNJ)”.
Comment: Line 267-269 – Just because these gene duplicates are under purifying selection, it does not mean that their divergence is driven by purifying selection—indeed they would be more diverged were it not for purifying selection. Purifying selection constrains divergence; it does not drive it. It would be more appropriate to say that their evolution was constrained by purifying selection. Indeed, high Ks values suggest that their divergence was driven by a high intrinsic mutation rate (and perhaps facilitated by drift) that was mitigated to some degree by purifying selection, as evidenced by the low Ka.
At the very least, this sentence: “Given the Ka/Ks ratios are all below one (from 0.0258 to 0.3751) (Table 2), their divergence is more likely to be driven by purifying selection.” should be changed to something like this: “Given the Ka/Ks ratios are all below one (from 0.0258 to 0.3751) (Table 2), their divergence is likely to be constrained by purifying selection.” and the authors can elaborate further if they feel so compelled (if they do so, I hope they keep my comments in mind).
Response: As suggested by the reviewer, “driven” was replaced by constrained in the revised version of manuscript.

Reviewer 2 Report
Comments and Suggestions for Authors
The manuscript entitled Genomic analysis of Carica papaya aquaporin genes provides insights into lineage-specific family evolution in Brassicales presented by authors Zou et al., seeks to gain insights into the AQP gene evolution and gene feature distribution pattern across Brassicales.
However, the lack of writing logic and low resolution figures (Figure 1 to Figure 5) make the manuscript unreadable. Therefore, the manuscript can not meet the quality of the Plants journal. Below, I listed few points.
1. The title “Genomic analysis of Carica papaya aquaporin genes provides insights into lineage-specific family evolution in Brassicales” does not fit to its content. The title should contain key gene AQP. In addition, I am wondering why using common name through the manuscript.
Line 13-34: The descriptions in this paragraph are in a mess. It is hard to interpret the meanings from these descriptions.
Line 16: “…relatively less than…” atypical writing style.
Line 38-40: citation. See “Mariana Chávez-Pesqueira and Juan Núñez-Farfán (2017), Domestication and Genetics of Papaya: A Review. Frontiers in ecology and evolution”. At least cite this manuscript.
Line 41-46: Lengthy. Please condense these sentences.
Line 47-63: These are descriptions without any conclusion. I can not agree with “Thereby, analysis of papaya gene families may improve our knowledge on lineage-specific evolution in Brassicales.” which was announced by the authors. In addition, the link between WGD and gene numbers is really weird.
Line 64-66: The rationale is weak. “…widely distributed…” is ambiguous.
Line 74-79: long sentence. I can not read or to understand.
Line 79-83: atypical writing. Consider “AQPs obtained from land plant lineages were clustered into seven phylogenetic groups including….”.
Line 92-93: I do not understand the sentence.
Line 95-97: unexpected conclusion sentence. I can not link this idea with previous description.
Line 107-111: ambiguous.
Line 146: The rationale needs to be described.
Line 155-156: rephrase.
Line 185-190: Why unevenly distributed AQPs observed in Sunset dataset preferred?
Line 198: I am wondering how to infer evolutionary relationship by using unrooted tree. More specifically, which clade diverged first in your phylogeny?
Line 260: Adding citation in the result section is atypical.
Line 267-269: move to “Discussion” section.
Line 516: “…singleton state…”? atypical usage.
Line 515-518: I do not understand why describe gene fate suddenly here.
Line 518-521: lack of connection.
Line 597-700: another result reporting. I do not see any significance of these sentences.
I am wondering why the Figure 6 was not discussed in the “discussion” section. In addition, IIIa to IIIc (Figure 6) were expressed at pretty low level. Can you provide any explanation?
Comments on the Quality of English LanguageMore extensive English editing is required.
Author Response
Response to Reviewer #2:
Comment: The manuscript entitled Genomic analysis of Carica papaya aquaporin genes provides insights into lineage-specific family evolution in Brassicales presented by authors Zou et al., seeks to gain insights into the AQP gene evolution and gene feature distribution pattern across Brassicales.
However, the lack of writing logic and low resolution figures (Figure 1 to Figure 5) make the manuscript unreadable. Therefore, the manuscript can not meet the quality of the Plants journal. Below, I listed few points.
Response: In the revised version of manuscript, all figures have been updated.
Comment: 1. The title “Genomic analysis of Carica papaya aquaporin genes provides insights into lineage-specific family evolution in Brassicales” does not fit to its content. The title should contain key gene AQP. In addition, I am wondering why using common name through the manuscript.
Response: In the revised version of manuscript, the title has been changed to “Analysis of Carica papaya informs lineage-specific evolution of the aquaporin (AQP) family in Brassicales”.
Comment: Line 13-34: The descriptions in this paragraph are in a mess. It is hard to interpret the meanings from these descriptions.
Response: It has been rewritten as “This study presents a first genome-wide identification and comparative analysis of the AQP gene family in papaya (Carica papaya L.), an economically and nutritionally important fruit tree of tropical and subtropical regions. A total of 29 CpAQPs were identified, which represent five subfamilies, i.e., nine plasma membrane intrinsic proteins (PIPs), eight tonoplast intrinsic proteins (TIPs), seven NOD26-like intrinsic proteins (NIPs), two X intrinsic proteins (XIPs), and three small basic intrinsic proteins (SIPs). Though the family is smaller than 35 members reported in arabidopsis, the presence of CpXIP genes and orthologs in horseradish tree and Bretschneidera sinensis implies that the complete loss of the XIP subfamily in arabidopsis is lineage-specific, sometime after its split with papaya but before Brassicaceae-Cleomaceae divergence. Best-reciprocal-hit-based sequence comparison of 530 AQP genes and synteny analyses revealed that CpAQP genes belong to 29 out of 61 identified orthogroups, and lineage-specific evolution was frequently observed in Brassicales”.
Comment: Line 16: “…relatively less than…” atypical writing style.
Response: It has been changed to “Though the family is relatively smaller than 35 members reported in arabidopsis”.
Comment: Line 38-40: citation. See “Mariana Chávez-Pesqueira and Juan Núñez-Farfán (2017), Domestication and Genetics of Papaya: A Review. Frontiers in ecology and evolution”. At least cite this manuscript.
Response: The paper has been cited as reference 1.
Comment: Line 41-46: Lengthy. Please condense these sentences.
Response: These sentences were condensed as “Papaya is sweet, flavorful, brightly colored, and uniquely rich in vitamin C and carotenoids, making it rank first on nutritional scores among 38 common fruits and also rank first among fruits consumed”.
Comment: Line 47-63: These are descriptions without any conclusion. I can not agree with “Thereby, analysis of papaya gene families may improve our knowledge on lineage-specific evolution in Brassicales.” which was announced by the authors. In addition, the link between WGD and gene numbers is really weird.
Response: The sentence was discarded and the conclusion was supplemented as “implying lineage-specific gene evolution and reflecting the occurrence of two additional whole-genome duplications (WGDs, known as At-β and At-α) followed by huge chromosomal rearrangement and massive gene loss occurred in arabidopsis after the split with papaya”.
Comment: Line 64-66: The rationale is weak. “…widely distributed…” is ambiguous.
Response: “distributed” has been changed to “found”.
Comment: Line 74-79: long sentence. I can not read or to understand.
Response: A “.” has been inserted after “[24,25]”.
Comment: Line 79-83: atypical writing. Consider “AQPs obtained from land plant lineages were clustered into seven phylogenetic groups including….”.
Response: The sentence has been changed to “according to sequence similarity, AQPs identified in land plant lineages were clustered into seven phylogenetic subfamilies including”.
Comment: Line 92-93: I do not understand the sentence.
Response: The sentence has been changed to “In cassava (Manihot esculenta), 13 out of 14 identified AQP repeats were shown to arise from the ρ recent WGD that was shared by rubber tree (Hevea brasiliensis) [30]”.
Comment: Line 95-97: unexpected conclusion sentence. I can not link this idea with previous description.
Response: The sentence has been changed to “whether XIPs are present in other Brassicales species beyond Brassicaceae and lineage-specific evolution patterns of the whole AQP gene family in Brassicales still need to be studied”.
Comment: Line 107-111: ambiguous.
Response: The sentence has been changed to “the presence of XIPs in papaya, horseradish tree, and B. sinensis but absent from spider flower imply that their loss in arabidopsis is lineage-specific, occurred sometime after the split with papaya but before Brassicaceae-Cleomaceae divergence”.
Comment: Line 146: The rationale needs to be described.
Response: The sentence has been changed to “To uncover the evolutionary rate of duplicate pairs, Ka (nonsynonymous substitution rate) and Ks (synonymous substitution rate) were calculated by codeml in the PAML package”.
Comment: Line 155-156: rephrase.
Response: The sentence has been changed to “Except for three novel groups identified in this study, classification of AQPs into subfamilies and groups was done as described before [35]”.
Comment: Line 185-190: Why unevenly distributed AQPs observed in Sunset dataset preferred?
Response: It is well known that the reliability of synteny analysis is highly dependent on the quality and integrity of the genome assembly, thereby the Sunset dataset was used for further analyses.
Comment: Line 198: I am wondering how to infer evolutionary relationship by using unrooted tree. More specifically, which clade diverged first in your phylogeny?
Response: In this study, we focused on the classification of AQPs but rather than which clade diverged first, thereby the unrooted tree was adopted.
Comment: Line 260: Adding citation in the result section is atypical.
Response: It has been revised.
Comment: Line 267-269: move to “Discussion” section.
Response: It has been revised.
Comment: Line 516: “…singleton state…”? atypical usage.
Response: It has been deleted.
Comment: Line 515-518: I do not understand why describe gene fate suddenly here.
Response: It has been deleted.
Comment: Line 518-521: lack of connection.
Response: We first introduced the order Brassicales and then discussed WGDs described in Brassicales.
Comment: Line 597-700: another result reporting. I do not see any significance of these sentences.
Response: These two paragraphs have been condensed.
Comment: I am wondering why the Figure 6 was not discussed in the “discussion” section. In addition, IIIa to IIIc (Figure 6) were expressed at pretty low level. Can you provide any explanation?
Response: Actually, Figure 6 was discussed in the last paragraph, which focused on highly abundant isoforms. IIIa to IIIc identified in this study includes most members from NIP, SIP, and XIP subfamilies, which were always characterized less expressed isoforms. Generally, NIPs and XIPs transport small solutes rather than water.

Reviewer 3 Report
Comments and Suggestions for Authors
In the manuscript plants-2612946 “Genomic analysis of Carica papaya aquaporin genes provides insights into lineage-specific family evolution in Brassicale”, authors identified 29 AQP genes in papaya and genome-wide abnalysis of them are performed in detail. Although there are no further functional studies of papaya AQP genes, the characterization of papaya AQP genes in the present study will facilitate further studies of AQP genes in papaya and other species. Consider the revision of the manuscript according to the points:
1) Authors emphasize lineage-specific loss of the XIP subfamily and several phylogenetic groups such as NIP2, NIP3, NIP8, and SIP3. However, there is no physiological insight of this phenomenon. Please add a possible discussion about why XIPs are present in papaya but not in Arabidopsis.
2) Probably because there are no wet experiments in the present manuscript, the result and discussion sections overlap too much. Please reduce the redundant (repeated) description.
3) No genome-wide analysis but papaya AQPs have been reported partially by Burns et al (Comparison of fruit texture and aquaporin gene expression in papaya “Khak Nual” cultivated under varying conditions. The Journal of Horticultural Science and Biotechnology (2023) https://doi.org/10.1080/14620316.2023.2206396). This paper should be cited and discussed.
4) I cannot distinguish the green and blue lines in Figure 1.
5) Why only “Ranunculaceae, A. coerula (Genome size (Mb); 306.5)” is bold in Table 3?
Comments on the Quality of English LanguageText length can be reduced.
Author Response
Response to Reviewer #3:
Comment: In the manuscript plants-2612946 “Genomic analysis of Carica papaya aquaporin genes provides insights into lineage-specific family evolution in Brassicale”, authors identified 29 AQP genes in papaya and genome-wide abnalysis of them are performed in detail. Although there are no further functional studies of papaya AQP genes, the characterization of papaya AQP genes in the present study will facilitate further studies of AQP genes in papaya and other species. Consider the revision of the manuscript according to the points:
Comment: 1) Authors emphasize lineage-specific loss of the XIP subfamily and several phylogenetic groups such as NIP2, NIP3, NIP8, and SIP3. However, there is no physiological insight of this phenomenon. Please add a possible discussion about why XIPs are present in papaya but not in Arabidopsis.
Response: No obvious phenotypic and physiological changes for the loss of XIPs may be due to functional redundancy with other subfamilies, e.g., NIP, which was proven to transport water, glycerol, urea, arsenite, selenite, boric acid, lactic acid, silicic acid, NH3, and H2O2.
Comment: 2) Probably because there are no wet experiments in the present manuscript, the result and discussion sections overlap too much. Please reduce the redundant (repeated) description.
Response: In the revised version of manuscript, we have tried to concise the manuscript.
Comment: 3) No genome-wide analysis but papaya AQPs have been reported partially by Burns et al (Comparison of fruit texture and aquaporin gene expression in papaya “Khak Nual” cultivated under varying conditions. The Journal of Horticultural Science and Biotechnology (2023) https://doi.org/10.1080/14620316.2023.2206396). This paper should be cited and discussed.
Response: In the revised version of manuscript, the paper has been cited and discussed. Thanks for the reminder by the reviewer. It is really a lately published paper.
Comment: 5) Why only “Ranunculaceae, A. coerula (Genome size (Mb); 306.5)” is bold in Table 3?
Response: In the revised version of manuscript, it has been revised and we apologize for the mistake.

Reviewer 4 Report
Comments and Suggestions for Authors
Zou et al.’s work bears striking resemblance to a series of authors' previous publications, especially the one in MDPI's 'Life' journal. The primary difference lies in the alteration of the gene family under investigation. Notably, no new sequence data have been provided to support the findings presented. This omission not only raises concerns but also represents a potential setback for the scientific community's pursuit of rigorous and well-substantiated research. In light of this, I strongly recommend that the authors consider incorporating comprehensive transgenic data to validate the functions attributed to these gene families in the future. Such an approach would not only bolster the credibility of their current work but also contribute to the advancement of our understanding without the proliferation of numerous gene family papers in the future. Therefore, I must reject this paper.
Comments on the Quality of English LanguageLanguage is clear.
Author Response
Response to Reviewer #4:
Comment: Zou et al.’s work bears striking resemblance to a series of authors' previous publications, especially the one in MDPI's 'Life' journal. The primary difference lies in the alteration of the gene family under investigation. Notably, no new sequence data have been provided to support the findings presented. This omission not only raises concerns but also represents a potential setback for the scientific community's pursuit of rigorous and well-substantiated research. In light of this, I strongly recommend that the authors consider incorporating comprehensive transgenic data to validate the functions attributed to these gene families in the future. Such an approach would not only bolster the credibility of their current work but also contribute to the advancement of our understanding without the proliferation of numerous gene family papers in the future. Therefore, I must reject this paper.
Response: We may have different opinions with the reviewer. Though it has been reported that the XIP subfamily is absent from Brassicaceae (e.g. the model plant arabidopsis), thus far, there is still a gap whether XIPs are present in other Brassicales families beyond Brassicaceae. In this study, we took advantage of representative plant species to address this issue, e.g., the basal eudicot Aquilegia coerulea (Ranunculaceae, Ranunculales), horseradish tree (Moringa oleifera, Moringaceae), Bretschneidera sinensis (Akaniaceae), spider flower (Tarenaya hassleriana, Cleomaceae), saltwater cress (Eutrema salsugineum, Brassicaceae), A. halleri (Brassicaceae), A. lyrata (Brassicaceae), and arabidopsis (Brassicaceae). Compared with other papers focusing on a gene family in a single species, this study aimed to uncover lineage-specific evolution of the AQP gene family in Brassicales. Significantly, our results showed that the complete loss of the XIP subfamily in arabidopsis is lineage-specific, occurred sometime after its split with papaya but before Brassicaceae-Cleomaceae divergence, since it is also absent from spider flower, C. violacea, and G. gynandraa, three Cleomaceae plants with available genomes. When compared with other eudicots such as poplar, castor bean, physic nut, rubber, cassava, we also found that the well-characterized NIP3 group was completely lost in Brassicales, lineage-specific loss of the NIP8 group in Brassicaceae occurred sometime before the divergence with Cleomaceae, and lineage-specific loss of NIP2 and SIP3 groups in Brassicaceae occurred sometime after the split with Cleomaceae. These findings provide valuable information for further studies of AQP genes in papaya and species beyond. Functional characterization of key genes is also of interest, however, we should keep in mind that one paper could not report all scientific issues.

Round 2
Reviewer 2 Report
Comments and Suggestions for Authors
I have no further questions.
Author Response
Thank you for your positive consideration and value suggestions for our manuscript.